# The Detectability of the Viral RNA in Blood and Cerebrospinal Fluid of Patients with Tick-Borne Encephalitis

**DOI:** 10.3390/ijms23169332

**Published:** 2022-08-19

**Authors:** Sambor Grygorczuk, Justyna Dunaj-Małyszko, Piotr Czupryna, Artur Sulik, Kacper Toczyłowski, Agnieszka Siemieniako-Werszko, Agnieszka Żebrowska, Sławomir Pancewicz, Anna Moniuszko-Malinowska

**Affiliations:** 1Department of the Infectious Diseases and Neuroinfections, Medical University in Białystok, Ul. Żurawia 14, 15-540 Białystok, Poland; 2Department of the Pediatric Infectious Diseases of the Medical University in Białystok, Ul. Jerzego Waszyngtona 17, 15-274 Białystok, Poland; 3University Hospital in Białystok, Ul. Żurawia 14, 15-540 Białystok, Poland; 4Regional Centre of Transfusion Medicine in Białystok, Ul. Marii Skłodowskiej-Curie 23, 15-950 Białystok, Poland

**Keywords:** tick-borne encephalitis, viremia, real-time PCR, molecular diagnostics

## Abstract

Background: The detection rate of viral RNA in tick-borne encephalitis (TBE) is low and variable between studies, and its diagnostic/prognostic potential is not well defined. We attempted to detect RNA of TBE virus (TBEV) in body fluids of TBE patients. Methods: We studied 98 adults and 12 children with TBEV infection, stratified by the disease phase and presentation. EDTA blood and cerebrospinal fluid (CSF) samples were obtained upon hospital admission. RNA was extracted from freshly obtained plasma, concentrated leukocyte-enriched CSF, and whole blood samples, and real time PCR was performed with a Rotor-Gene Q thermocycler. Results: TBEV RNA was detected in (1) plasma of one (of the two studied) adult patients with an abortive infection, (2) plasma of two (of the two studied) adults in the peripheral phase of TBE, and (3) plasma and blood of an adult in the neurologic phase of TBE presenting as meningoencephalomyelitis. No CSF samples were TBEV RNA-positive. Conclusions: The detection of TBEV RNA in blood might be diagnostic in the peripheral phase of TBE. The lack of TBEV RNA in the CSF cellular fraction speaks against TBEV influx into the central nervous system with infiltrating leukocytes and is consistent with a relatively low intrathecal viral burden.

## 1. Introduction

Tick-borne encephalitis (TBE) is an acute central nervous system (CNS) infectious disease caused by tick-borne encephalitis virus (TBEV), a *Flavivirus* transmitted by *Ixodes* ticks. The disease is endemic in most of the moderate climate zone in Europe and Asia, and several thousand cases are reported annually [1,2,3,4]. In Poland, 200–300 cases are diagnosed annually, mostly in the northeast of the country, with a morbidity reaching 9.16 per 100,000 inhabitants in Podlasie province in 2019 [5]. Only the European TBEV subtype has been found in Poland, and little sequence variability between the isolates has been reported [6]. TBE progresses from the phase of peripheral infection, through the blood–brain barrier (BBB) crossing, to the central nervous system (CNS) infection [7,8,9]. The peripheral viremia is apparently a first step towards neuroinfection, but its level does not correlate with the severity of the subsequent neurologic disease [10]. The route of BBB crossing involves infection of and replication within the endothelial cells, but additional virus entry within migrating leukocytes, as observed in animal models of *Flavivirus* encephalitis, may contribute to the CNS viral burden [11,12]. Intrathecally, TBEV infects neurons and astrocytes, but the local viral load tends to be low relative to the extent of the accompanying inflammation [7,9,10,13,14].

Clinically, the peripheral infection is reflected by an initial, flu-like phase of the disease, in which the symptoms from the CNS are absent, and the overall clinical presentation is rather unspecific [1,15]. In the rare abortive cases, the disease may not progress further and remains limited to the periphery [16]. The majority of TBE patients, however, progress to a second, neurologic phase [15,16], which may greatly vary in severity and presentation, from an uncomplicated meningitis to a life-threatening encephalitis and/or myelitis. The peripheral and neurologic phases may be clinically distinct, separated by a period of remission, or the disease may take a more rapidly progressing and clinically monophasic course [1,2,3,4,17].

TBEV RNA is rarely detectable in the body fluids of TBE patients, with the positive results most often reported very early in the course of the infection or in immunocompromised patients [9,10,18,19]. Accordingly, the molecular methods are not considered as routine tools in TBE diagnosis, and the presence of the specific anti-TBEV IgM antibodies is used for TBE confirmation [20]. The limitations of this approach include a potential cross-reactivity in the areas where different *Flavivirus* species co-circulate or in travelers [21,22] and the prolonged (up to 6–8 months) detectability of the specific IgM after a TBEV infection or an anti-TBE vaccination, raising the possibility of false-positive results [23]. Moreover, the peripheral phase of TBE is generally seronegative, which, together with an unspecific clinical presentation, makes its diagnosis, and, consequently, the prediction of the neurologic phase, highly problematic [19,22]. Although no causative treatment of TBE exists, early diagnosis could be beneficial by shortening the time to hospitalization and to appropriate supportive treatment. Thus, the clinical use of TBEV RNA detection has been suggested in seronegative patients with an epidemiologic history and presentation consistent with early TBEV disease [9,19]. However, better knowledge of TBEV RNA dynamics during TBE, especially in relation to serology, is needed to assess its potential clinical usefulness.

We attempted to detect RNA of TBE virus in blood and cerebrospinal fluid (CSF) of a group of well-characterized TBE patients to assess its presence, dynamics, factors related to its detection, and potential diagnostic and prognostic value in clinical practice and in studies of the pathogenesis of TBE.

## 2. Results

### 2.1. Study Groups

There were 95 adult patients studied during a neurologic TBEV infection, including 73 from whom paired CSF samples were available. There were 63 males and 32 females, aged 19–79 years (median 47). Eighteen patients reported significant systemic comorbidities (cardio-vascular disease, diabetes mellitus, hyper- and hypothyroidism, hypoparathyroidism, chronic renal disease, rheumatoid arthritis), but none had active malignancy, HIV infection, immunosuppressive treatment, or other potential causes of severe immunosuppression. Forty-eight patients had a biphasic presentation (the disease duration prior to sample collection 7–30 days, median 19; neurologic phase duration 2–14 days, median 6), and 47 had a clinically monophasic disease (duration 1–21 days, median 6). Fifty four (56.8%) had uncomplicated meningitis (M), 35 (36.8%) meningoencephalitis (ME, classified as mild in 22, moderate in 11, and severe in 2), and 6 (6.3%) meningoencephalomyelitis (MEM, mild in 2, moderately severe in 2, severe in 2). No patient died, required life-supporting therapy, or had life-threatening complications. All were seropositive towards TBEV at the time of the sample collection. In 9 patients, follow-up (convalescent) samples (paired blood and CSF) were available. In one patient, a woman 59 years old with M, additional blood and CSF sample pairs from the peripheral phase of the disease were available; the patient was seronegative and had normal CSF when these samples were collected.

The blood samples were obtained during the peripheral phase from three additional adult patients with a final diagnosis of a TBEV infection: a seronegative woman who later progressed to meningitis and seroconverted (blood only), a seronegative woman in the 3rd day of an abortive infection (paired blood and CSF), and an IgM-seropositive woman in the 7th day of an abortive infection (blood only).

The pediatric TBE group consisted of 12 patients, 10 male and 2 female, aged 6–17 years (median 9 years); 7 with M and 5 with mild to moderately severe ME; and all seropositive towards TBEV. The CSF samples were available in 11 of these patients.

There were 17 patients with non-TBE aseptic meningitis (AM), 11 male and 6 female, aged 19–84 years (median 33), 14 with meningitis and 3 with meningoencephalitis. CSF samples were available from 15 of them.

The healthy blood donor group included 300 subjects, from 18 to 63 years old, 231 males and 69 females. This large cohort was used to validate the specificity of the method in persons with very low probabilities of active infections.

### 2.2. TBEV RNA Detection in Plasma and CSF

#### 2.2.1. Controls

There were no positive results, neither in plasma samples from healthy controls, nor in whole blood, plasma, and CSF samples from non-TBE meningitis patients, confidently confirming the specificity of the procedure.

#### 2.2.2. Blood of TBE Patients

Only one adult TBE patient had TBEV RNA detected in plasma and full blood during the neurologic phase of the disease—a woman 68 years old, immunocompetent, anti-TBEV IgM and IgG positive in serum and CSF, with a biphasic disease lasting 19 days since the onset and 8 days since the beginning of the neurologic phase. She presented with MEM of moderate severity, which was a rare presentation in the group as a whole. Additionally, her presentation, outcome, blood, and CSF inflammatory parameters were within the range found in the remaining patients.

Of four patients studied during the peripheral infection, three were positive for TBEV RNA in plasma but negative in full blood: two patients examined during the peripheral phase of the biphasic infection and one in the 3rd day of the abortive infection. The patient studied on the 7th day of the abortive disease after IgM seroconversion was TBEV RNA-negative.

No pediatric patient had TBEV-RNA detected either in plasma or whole blood samples.

No follow-up sample was rtPCR positive.

#### 2.2.3. CSF of TBE Patients

All the CSF samples from TBE patients were TBEV RNA-negative, irrespective of age, time since onset, and presentation, and including the samples paired with the TBEV RNA-positive blood.

An overview of the results is shown in Table 1, and the basic data of the patients with the positive results are summed up in Table 2.

## 3. Discussion

We confirmed previous observations on the low prevalence of TBEV RNA in the body fluids of patients with TBE [9,22,24,25]. The other authors reported either low rates of detection [9,26] or non-detection of TBEV RNA [24,25] in the TBE neurologic phase. The lack of a significant viremia is expected at this stage of the disease, in agreement with the animal models of *Flavivirus* encephalitis [8]. More surprisingly, TBEV RNA is typically undetectable in CSF, in spite of the simultaneous pleocytosis and ongoing CNS inflammation [21,24,25,26,27,28]. Originally, Puchhammer-Stöckl et al. detected TBEV in only 1 of 105 TBE CSF samples [26]. Saksida et al. reported 10% sensitivity of the rtPCR of CSF [9], which has not been repeated in further studies [22,24,25], including our own. This has important consequences for TBE diagnostics, which must rely on serologic confirmation [20,26]. Small viral loads relative to the intense inflammatory response and the clinical severity have implications for the pathogenesis of the TBE neurologic phase as well, suggesting that host-dependent mechanisms contribute to the intrathecal inflammation. Moreover, the apparent lack of TBEV in CSF leukocytes means they are not an important vehicle of CNS infiltration by TBEV, at least during the clinically overt neurologic phase.

The profound immunosuppression, which is frequently accompanied by a severe encephalitic presentation of TBE, may favor TBEV RNA detection [18,29]. For example, Caracciolo et al. described a blood PCR remaining positive for 74 days in a fatal TBE case in a patient undergoing chemotherapy. Our study cohort did not include transplant recipients, patients with known active neoplasms, HIV-positive, or other deeply immunosuppressed patients, which did not allow us to confirm this observation. However, it did include patients with diabetes mellitus and other hormonal disorders, rheumatoid arthritis, and cardiac insufficiency, but we did not detect TBEV RNA in any of them. We infer that the moderate immune impairment related to common chronic comorbidities does not increase the rate of TBEV RNA detection notably. We also did not observe any association of TBEV RNA detection with encephalitis or altered mental status. However, the only positive neurologic phase result was obtained in a patient in a relatively advanced age (at the 90 percentile of the cohort) and presenting with a rare, clinically distinct, and often severe myelitic form of TBE. While this single detection does not allow for any definite conclusions, the association of TBEV RNA in blood with age and/or TBE presenting as MEM is feasible and could be verified by further studies. Interestingly, this patient presented with the lowest ct value in plasma in the whole study group and was the only one with a simultaneous positive result in a full blood sample, suggesting a higher viral load than in the positive peripheral phase samples.

In most of the previous studies, there was a tendency for a better detectability of TBEV RNA in the whole blood [9,22,29] and blood erythrocyte fraction [18] compared to plasma. The same phenomenon was previously described for another neurotropic *Flavivirus*, the West Nile virus (WNV), and was suggested to result from the virion–antibody complex affinity to erythrocyte membranes [30]. In our study, the whole blood analysis did not reveal any additional detections and, on the contrary, TBEV RNA was not detected in whole blood from three patients with positive plasma results. This disappointing result could be at least partially attributed to the freezing and thawing cycle of the whole blood samples, which, unlike plasma, were stored and analyzed after the conclusion of the sample collection period.

Interestingly, TBEV RNA has been previously detected in urine simultaneously with or even after the end of the detectable viremia, which is also analogous to a tendency known for WNV and if verified could offer another diagnostic possibility [18,21,22,31]. Further research should lead to optimizing the choice of samples for TBEV RNA detection and to defining patient groups with a higher prior probability of positive results. However, the main factor determining the detectability of TBEV RNA seems to be the time since the disease onset, with overwhelmingly higher rates of positive results in the peripheral as compared to the neurologic phase, and in seronegative as compared to seropositive patients. In the Saksida et al. study, the fraction of TBEV-positive serum samples dropped by 1/3 after the seroconversion in IgM class and to 3% after IgG seroconversion. In a later study, Saksida et al. detected TBEV RNA in sera of all the studied patients in the peripheral phase, but in none studied in the neurologic phase of TBE [10]. Our results are in strong agreement with those of Saksida et al.—three out of four TBEV RNA detections in plasma were achieved in the seronegative patients (in the peripheral phase or early in the abortive infection), while all the negative results occurred in the seropositive patients (in the neurologic phase or in the abortive infection after seroconversion). The only exception was a single TBEV RNA-positive patient with MEM discussed above.

In the peripheral phase, the clinical symptoms do not allow for the reliable differentiation between the incipient TBE and other febrile tick-borne infections (anaplasmosis, ehrlichioses, spotted fever group rickettsioses, borrelial recurrent fever group infections, etc., depending on the local epidemiology) [32]. The most common laboratory findings in the peripheral phase of TBE (leucopenia, thrombocytopenia, increased transaminase activity) are unspecific and do not differentiate it from febrile tick-borne bacterial infections either [1,16,32]. The TBEV RNA detection is the only method offering a possibility of a specific and reliable diagnostics in this stage of the disease. For example, Bogovič et al. were recently able to retrospectively identify 88 patients in the peripheral phase of TBE within the large population of febrile tick-exposed patients seen in the highly endemic TBE area in Slovenia based on positive PCR results, which points to the diagnostic potential of this method [15].

As no causative TBE treatment is currently available, the early diagnosis made possible by rtPCR would not lead to a specific cure or the prevention of the neurologic disease. However, it could facilitate the decision about early hospitalization and supportive care and prevent an ineffective empirical antibiotic treatment. Apart from any clinical benefits, a prompt identification of patients in the peripheral phase of TBE would greatly facilitate the clinical studies on this relatively poorly characterized stage of the disease. The results of Bogovič et al., who were able to assess longitudinally a number of immune parameters in their patient cohort and pinpoint intriguingly different immune response patterns between the peripheral and neurologic phase, highlight the research potential of diagnosing TBE in this early stage [15]. According to Saksida et al., the peripheral phase viral load by itself is not correlated with CSF parameters and clinical presentation of the neurologic phase, which suggests it is not a factor determining the neuroinvasion and is not a good prognostic marker [10]. However, further studies could reveal other early clinical or laboratory parameters with a higher prognostic value. Optimally, patients who will or will not progress to the neurologic phase could be identified and the risk of the severe neurologic presentation assessed. The peripheral phase of TBEV infection, with active viral replication at the periphery, would also be an optimal period for any attempts at anti-viral treatment.

## 4. Materials and Methods

### 4.1. Patients

Patients hospitalized in the Department of the Infectious Diseases and Neuroinfections and the Department of the Pediatric Infectious Diseases of the Medical University in Białystok in 2019–2022 with a suspicion of TBE were recruited to the study. The patients included in the TBE group (1) had a history of a tick bite or an exposition to ticks in an endemic area within 3 weeks before the onset of symptoms, (2) presented with an acute febrile disease with symptoms suggestive of meningitis and/or encephalitis, (3) had a CSF pleocytosis ≥15/µL, and (4) either had specific anti-TBEV IgM antibodies detected in serum and/or in CSF on admission or seroconverted on follow-up, fulfilling the European criteria for the confirmed TBE case [20]. The patients fulfilling criteria (2) and (3) but seronegative towards TBEV both on admission and on follow-up were diagnosed with aseptic meningitis (AM) and formed a meningitis reference group. The patients with an acute febrile infection beginning within 2 weeks after a tick bite, with no CSF pleocytosis not fulfilling criterion (3) but without a likely alternative diagnosis, were tentatively recruited into the study cohort as possibly presenting with a peripheral phase of TBE or an abortive TBEV infection.

The TBE patients with a meningeal syndrome but neither neurologic abnormalities nor altered consciousness were classified as having uncomplicated meningitis (M), and patients with an altered mental status and/or any objective neurologic symptoms—as having meningoencephalitis (ME) or meningoencephalomyelitis (MEM), which was further classified as mild (minor neurologic abnormalities including sensory deficits, pathologic reflexes, tremor, unstable gait), moderately severe (lethargy, focal paresis, cerebellar syndrome), or severe (disorientation or loss of consciousness, multiple and/or severe focal deficits, seizures). The patients reporting an initial flu-like disease followed by an improvement and a relapse with CNS involvement were classified as having a biphasic presentation, and the remaining ones as having a monophasic presentation.

The control group consisted of healthy blood donors.

The blood and CSF samples from the hospitalized patients were obtained together with the material for clinically indicated examinations, typically within the first day of the hospital stay. The control blood samples were obtained from the blood donors directly before a donation.

The patients and healthy controls gave informed written consent for inclusion. The study was approved by the Ethics Committee of the Medical University in Białystok (approval no R-I-002/308/2019).

### 4.2. Laboratory Examinations

#### 4.2.1. Diagnostic Examinations

The basic laboratory examinations were performed in the hospital laboratory on the blood and CSF samples obtained on admission, as a part of the routine diagnostics. Anti-TBEV IgM and IgG antibodies were detected with Enzygnost Anti-TBE/FSME IgM and Anti-TBE/FSME IgG kits from Siemens (Munich, Germany) following the standard procedure, in serum and CSF samples obtained on admission and on follow-up in the originally seronegative patients.

#### 4.2.2. Sample Processing

A 1.4 mL sample of venous blood was drawn into anticoagulant-coated PAXgene Blood RNA tube from Qiagen (Hilden, Germany) and stored for 2–4 h at 2–6 °C. Before the RNA extraction, it was divided into 2 aliquots, one of which was centrifuged for plasma separation at 1500 RPMI for 10 min at 4 °C, and the other of which was retained for extraction from the whole blood. A 1 mL CSF sample was obtained during a diagnostic lumbar puncture into a sterile vial and was stored together with blood and processed simultaneously (in individual patients, CSF was not available for the study purposes because of too small a volume collected during the lumbar puncture). Before the RNA extraction, CSF was centrifuged at 8000 RPMI for 5 min to separate the concentrated cellular fraction. The RNA extraction from CSF and plasma was performed on fresh samples, while whole blood samples were frozen to −20 °C and analyzed at the end of the sample collection period. Frozen samples were thawed for at least 2 h at room temperature before RNA extraction.

#### 4.2.3. RNA Extraction

The RNA extraction from plasma, thawed full blood, and concentrated CSF was performed with a Blood and Tissue RNA Mini Kit (Qiagen) following the manufacturer’s instructions. The procedure includes addition of DNase to exclude any possibility of the RNA contamination with DNA. RNA quality was confirmed by using internal control added just before the RNA extraction step—to each 200 µL of cerebrospinal fluid was added 10 µL of Internal Control STI-87-rec (IC).

#### 4.2.4. Real-Time PCR

Real-time PCR (rtPCR) for TBEV RNA detection was performed with an Amplisens TBE-FRT PCR kit, based on the amplification of the pathogen genome specific region (target gene—C gene), with specific primers (sequences of starters are covered by the Producer’s patent). According to the product characteristics card, the analytical specificity of the Amplisens TBE-FRT PCR kit was ensured by the selection of specific primers and probes, as well as stringent reaction conditions, and the primers and probes were checked for possible homologies to all sequences published in gene banks by sequence comparison analysis, and the clinical specificity was confirmed in laboratory clinical trials.

The Amplisens TBE-FRT PCR kit guarantees reduction of nonspecific reaction by using Hot Start technology. This is possible because of the separation of nucleotides and chemically modified polymerase (TaqF), which is activated by heating at 95 °C for 15 min. Following the Producer’s instructions for a single probe reaction, 10 µL of extracted RNA was added to 15 µL of the mixture of 10 µL of RT-PCR-mix-1-FEP/FRT TBE, 5 µL of RT-PCR-mix-2-FEP/FRT, 0.5 µL of polymerase (TaqF), 0.25 µL of TM-revertase (MMIv), and 0.25 µL of RT-G-mix-2 to obtain a final volume of 25 µL. Amplification was performed in a RotorGeneQ Thermocycler (Qiagen, Germany) in the following conditions: hold step—50 °C for 30 min, hold step 2—95 °C for 15 min, cycling step—5 cycles (95 °C for 10 s, 65 °C for 45 s, 72 °C for 15 s), cycling step 2—45 cycles (95 °C for 10 s, 60 °C for 45 s—fluorescence acquiring, 72 °C for 15 s). The fluorescent signal was detected in the channels for the FAM and JOE fluorophores.

The internal controls for each sample, positive and negative controls, prepared by the manufacturer, were run simultaneously with the study samples.

## 5. Conclusions

The rtPCR for TBEV RNA has a very low sensitivity in patients in the neurologic phase of TBE. At this stage, TBEV RNA detection is possible in a minority of patients, having no proven diagnostic or prognostic significance.

We suggest a possible association of the TBEV viremia persisting into the neurologic phase with the myelitic presentation of TBE, which could be verified by further studies.

The lack of TBEV RNA in the cellular CSF fraction contradicts the substantial TBEV presence in leukocytes infiltrating CNS and favors other entry routes of TBEV into the CNS. It is also consistent with low viral titers within the CNS and a relatively large role of immune-mediated pathology.

The detection of TBEV RNA in blood might enable the diagnosis of the peripheral phase of TBE before seroconversion, with a potential for not only improving clinical practice but also facilitating research on TBE pathogenesis and treatment.

## Figures and Tables

**Table 1 ijms-23-09332-t001:** The proportion of the positive results of the rtPCR test for TBEV RNA in the study cohort. The number and percentage (in parentheses) of the positive samples per patient group and the type of diagnostic material.

Presentation and Stage of the Disease	Sample Type		
	Whole Blood	Plasma	CSF
Abortive infection	0/2 (0%)	1/2 (50%)	0/1 (0%)
First phase of a biphasic infection	0/2 (0%)	2/2 (100%)	0/1 (0%)
Neurologic phase—adults	1/95 (1%)	1/95 (1%)	0/73 (0%)
Biphasic infection	1/48 (2%)	1/48 (2%)	0/39 (0%)
Monophasic infection	0/41 (0%)	0/41 (0%)	0/34 (0%)
Meningitis	0/54 (0%)	0/54 (0%)	0/45 (0%)
Meningoencephalitis	0/35 (0%)	0/35 (0%)	0/24 (0%)
Meningoencephalomyelitis/radiculitis	1/6 (17%)	1/6 (17%)	0/4 (0%)
Convalescent period—adults	ND	0/9 (0%)	0/9 (0%)
Neurologic phase—children	ND	0/12 (0%)	0/12 (0%)

CSF—cerebrospinal fluid; ND—not done.

**Table 2 ijms-23-09332-t002:** The basic clinical and laboratory parameters of patients with positive results of a plasma rtPCR test for TBEV RNA.

No.	Sex	Age	Comorbidities	Presentation ^1^	Phase ^2^	Main Symptoms ^2^	Time Since Onset ^2^	Anti-TBEV IgM in Serum ^2^	TBEV RNA in Plasma (ct Value)	TBEV RNA in Whole Blood (ct Value)	TBEV RNA in CSF ^2^	Follow Up TBEV RNA
1	M	68	hypertension, hyperuricaemia, Lyme borreliosis	MEM	neurologic	headache, vertigo, neck stiffness, flaccid paresis	19 days; neurologic phase—8 days	negative	30.6	32.0	negative	ND
2	F	51	none	meningitis	peripheral	fever, arthralgia, headache	5 days	negative	32.6	negative	ND	negative plasma and CSF on day 17 (6th day of the neurologic phase)
3	F	59	possible Lyme borreliosis	meningitis	peripheral	fever, headache, maculopapular rash	4 days	negative	32.5	negative	negative	ND
4	F	31	none	abortive infection	peripheral	headache, myalgia, nausea, neck stiffness	3 days	negative	35.2	negative	negative	ND

^1^ Final diagnosis on discharge. ^2^ At the time of the TBEV RNA-positive plasma sample collection. M—male; F—female; MEM—meningoencephalomyelitis; ND—not done.

## Data Availability

The datasets used and/or analyzed during the current study are available from the corresponding author upon reasonable request.

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
