# Peer review of "The Detectability of the Viral RNA in Blood and Cerebrospinal Fluid of Patients with Tick-Borne Encephalitis"

_ijms, 2022, doi:10.3390/ijms23169332_

Round 1
Reviewer 1 Report
The manuscript by Grygorczuk et al. entitled “The detectability of the viral RNA in blood and cerebrospinal fluid of patients with tick-borne encephalitis” showed TBEV RNA detections in the clinical samples of blood and CSF to assess the diagnosis methods. The trial using clinical samples of TBE is valuable and important, and will provide useful information. However, this reviewer raises some concerns as enumerated below:
Major concern
In the Introduction section (line74-77), authors raise a concern of current TBE diagnosis including TBEV detection, and mention that the purpose of this study is to assess TBEV RNA detection related with the dynamics, factors, and potential diagnostic and prognostic value. However, results of this study did not support to resolve the raised concern. In addition, it is not clear what is novel points of the results of this study compared with previous studies particularly TBEV RNA detection in the clinical samples. Authors should clearly show the advantage of the results of this study compared with previous studies.
Minor concerns
1. Detailed methods of real-time RT-PCR of RNA detection such as the information of primer sequence and PCR reaction condition should be described in the methods section for readers to understand and refer.
2. Results of RNA detections are just shown by positive or negative. Data should be shown by the detection level such as copy numbers or ct values.
3. Description of discussion section is too redundant. It should be described based on the result of this study.
4. It is better to subdivide the section of 4.2 laboratory examination, for example, clinical samples, RNA extraction, real-time RT-PCR.
5. Abbreviation of meningitis (M) (line 268), meningoencephalitis (ME) (line 270) and meningoencephalomyelitis (MEM) (line 270) should be described in line 89 and 90.
Author Response
Major concern:
Indeed, our conclusions were limited by a very high fraction of negative results, comparable but still higher than in most of the previous studies. Regarding the typical, seropositive and neurologic presentation of TBE, we mainly confirm the results of the previous studies in a relatively large and well clinically characterized patient population – we show that the blood and CSF rtPCR is almost universally negative, including in different patient subgroups (elderly, chronically ill, patients with a severe encephalitis presentation), with important consequences for TBE diagnostics and pathogenesis research we mention in paragraph 1 of the Discussion.
However, our results also suggest the directions of a further study: in patients with a peripheral infection/before seroconversion (which has been suggested before and confirmed by our results) and with a myelitic presentation (which is novel, but needs confirmation).
We have rewritten paragraphs 2-4 to highlight more our results and their potential significance (as described above) and less the literature data. For example, we much limit the discussion of TBE in immunodeficiency and TBEV detections in different body fluid, to which we have not contributed in our study, but highlight more the detection of TBEV viremia in a MEM patient. However, the general order of the paragraphs remains as previously.
In Conclusions, the first paragraph has been rewritten for clarity and one additional sentence was added after it concluding our results in a patient with myelitis.
We have slightly revised Abstract in l.25-26, to highlight the myelitis patient more
Minor concerns:
- This information has been provided – it forms the body of the new subsection 4.2.4., replacing the previous, much shorter descritption.
- The ct values were given in table 2
- As stated above, the paragraphs 2-4 of the Discussion were reviewed and shortened to concentrate more on our own results and less on literature data (also in response to the Major concern).
- The section 4.2 has been divided into four subsections, as suggested.
- The full terms “meningitis”, “meningoencephalits”, “meningoencephalomyelits” were introduced in ll. 89-91
Reviewer 2 Report
Review of manuscript ijms-1817551
The manuscript under review is devoted to the detectability of the viral RNA in body fluids of patients with tick-borne encephalitis (TBE) with the aim to clarify diagnostic and prognostic potential of this method. Whole blood, serum and cerebrospinal fluid were examined for the presence of viral RNA in 98 adults and 12 children. Viral RNA was detected in plasma or whole blood only in 4 adult patients, in 3 ones in peripheral phase and in one in neurological phase (meningoencephalomyelitis) of TBE. This is a nice clinical study performed with bigger study cohort compared to previously published article dealt with same topic by Saksida et al (Saksida, 2006) and it brings the confirmation about viral RNA detection in blood mainly in peripheral phase of TBE and therefore I believe it possess a significance in the field. Study is designed well, results are clearly presented and discussion is very well written. Patients classifications (study cohort) are sufficiently described according the symptoms. Overall the study is of quite high interest because of providing proof about the detectability of viral RNA in early phase of TBE.
Comments:
1. The complete lack of TBEV RNA in CSF is quite surprising therefore information about RNA quality and amounts in CSF samples should be provided.
2. Blood was stored up to 72 hours in PAXgene Blood DNA tubes before proceeding to the isolation of RNA. Why PAXgene Blood RNA tubes were not used? As low detection of viral RNA could be due to low quality of isolated RNA, it is important to verify it. Was the quality of isolated RNA verified? What internal control for each sample (whole blood and plasma) was used? This information is important and should be mentioned in Materials and Methods.
Minor comments:
3. Table 2 requires formatting.
4. First sentence in conclusion requires a correction to make it clear.
5. Formal issues: Replace first usage of ‘CSF’ in abstract by term ‘cerebrospinal fluid.
6. On page 3, where ‘AM’ first appears explain what this abbreviation means.
Author Response
- The lack of RNA detections in CSF was disappointing, but it is not totally unexpected given very low rates of positive results in the other studies we discuss.
We agree with the Reviewer that there is a need to quality and amount of obtained RNA. In our study, RNA quality was confirmed by using internal control added just before RNA extraction step – to each 200 µl of cerebrospinal fluid was added 10 µl of Internal Control STI-87-rec (IC). The same extraction procedure as with CSF was performed with blood samples. RNA extraction was performed with good quality Qiagen kit and with DNase to exclude any possibilities of contamination RNA with DNA.
- In practice, blood was in storage for up to 2-4 h before extraction, in 2-6° C (). (The 72 hour period was set a priori as a maximum for a sample to be acceptable – in reality we managed to collect the blood samples in the morning and process them on the same day; for CSF, it could have been occasionally longer as lumbar punctures were performed as urgent procedures at different times of the day, but also << 24 hours in a vast majority of cases – we have corrected this in the text). Blood was collected into PAXgene RNA tubes (misprint appeared in the manuscript – has been corrected). Before extraction 10 µl of Internal Control STI-87-rec (IC) was added to whole blood (200 µl ) and serum samples (200 µl). RNA extraction was performed with good quality Qiagen kit and with DNase to exclude any possibilities of contamination RNA with DNA.
The information on internal control and DNase inclusion was added in the Methods (in the subsection “4.2.3. RNA extraction” created from the 3rd paragraph of the previous 4.2. section)
- Table 2 formatting has been improved
- The first sentence of the Conclusions section was rewritten
- “cerebrospinal fluid” has been added (l. 20)
- AM (aseptic meningitis) abbreviation was explained (l. 107)
Reviewer 3 Report
In the present study, the authors attempted to detect RNA of TBE virus (TBEV) in body fluids of 110 TBE patients. The manuscript is well written and provides valuable data on the possibilities of applying rtPCR as a diagnostic method in TBE patients.
Minor comments:
Line 19: Consider changing: …samples were obtained on admission to hospital… to: samples were obtained upon hospital admission.
Line 21 and 309: ..Rotor-Gene Q termocycler. to Rotor-Gene Q thermocycler
Line 35: .. cases reported annually. to ..cases are reported annually.
Line 62: ..travelers to travellers.
Line 86: .. Forty eight to Forty-eight
Line 89-90: Since the terms are mentioned for the first time in the manuscript at this point, use the full name and not the abbreviations (M, ME, MEM)
Line 307: ..each RNA extracts..to ..each RNA extract
Author Response
Line 19: text has been changed as suggested
Line 21 and 309 (now l. 321): ‘thermocycler’ spelling corrected
Line 35: style changed as suggested (‘cases are reported annually’)
Line 62: ‘travellers’ spelling corrected
Line 86: “Forty eight” changed to “Forty-eight”
Line 89-91: corrected, has been already addressed in p.5 of the response to Reviewer 1
Line 307: ‘extracts’ – all this text has been changed in response to the other Reviewers’ remarks